# Implications of GNSS-Inferred Tropopause Altitude Associated with Terrestrial Gamma-ray Flashes

Tao Xian [1,2] , Gaopeng Lu [1,3,4,*], Hongbo Zhang [5], Yongping Wang [1], Shaolin Xiong [6], Qibin Yi [6,7], Jing Yang [1] and Fanchao Lyu [8]

1   School of Earth and Space Science, University of Science and Technology of China, Hefei 230026, China; xiantao@mail.ustc.edu.cn (T.X.); wyp8966@mail.ustc.edu.cn (Y.W.); yangjing@mail.ustc.edu.cn (J.Y.)
2   Department of Mechanics and Aerospace Engineering, Southern University of Science and Technology, Shenzhen 518055, China
3   Collaborative Innovation Center on Forecast and Evaluation of Meteorological Disasters, Nanjing University of Information Science and Technology, Hefei 210044, China
4   Key Laboratory of Atmospheric Optics, Anhui Institute of Optics and Fine Mechanics, HFIPS, Chinese Academy of Sciences, Hefei 230031, China
5   Key Laboratory of Middle Atmosphere and Global Environment Observation (LAGEO), Institute of Atmospheric Physics, Chinese Academy of Sciences, Beijing 100029, China; zhanghb@mail.iap.ac.cn
6   Key Laboratory of Particle Astrophysics, Institute of High Energy Physics, Institute of Atmospheric Physics, Chinese Academy of Sciences, Beijing 100029, China; xiongsl@ihep.ac.cn (S.X.); yiqibin@ihec.cas.cn (Q.Y.)
7   School of Physics and Optoelectronics, Xiangtan University, Yuhu District, Xiangtan 411105, China
8   Nanjing Joint Institute for Atmospheric Sciences, Chinese Academy of Meteorological Sciences, Nanjing 210000, China; lyufc@cma.gov.cn
*   Correspondence: gplu@ustc.edu.cn

**Abstract:** The thermal structure of the environmental atmosphere associated with Terrestrial Gamma-ray Flashes (TGFs) is investigated with the combined observations from several detectors (FERMI, RHESSI, and Insight-HXMT) and GNSS-RO (SAC-C, COSMIC, GRACE, TerraSAR-X, and MetOp-A). The geographic distributions of TGF-related tropopause altitude and climatology are similar. The regional TGF-related tropopause altitude in Africa and the Caribbean Sea is 0.1–0.4 km lower than the climatology, whereas that in Asia is 0.1–0.2 km higher. Most of the TGF-related tropopause altitudes are slightly higher than the climatology, while some of them have a slightly negative bias. The subtropical TGF-producing thunderstorms are warmer in the troposphere and have a colder and higher tropopause over land than the ocean. There is no significant land–ocean difference in the thermal structure for the tropical TGF-producing thunderstorms. The TGF-producing thunderstorms have a cold anomaly in the middle and upper troposphere and have stronger anomalies than the deep convection found in previous studies.

**Keywords:** Terrestrial Gamma-ray Flashes (TGFs); tropopause; deep convection; GNSS-radio occultation

## 1. Introduction

Terrestrial Gamma-ray Flashes (TGFs) are sub-millisecond bursts of energetic MeV gamma-ray photons produced in the troposphere that can be detected on the low-Earth orbit [1–5] or even at ground level [6,7]. It is known that TGFs are forged inside thunderstorms and related to the initial upward leader development of intra-cloud (IC) lightning flashes [8–11], or downward propagating negative leaders [12–14]. However, the characteristic charge structure of TGF-producing thunderstorms and the associated convection properties remain an open question to be investigated. The authors of [15] found a preference for TGF-producing thunderstorms for larger values of surface convective available potential energy (CAPE), whereas the authors of [16] recently found no difference in CAPE

between tropical TGF-producing thunderstorms and local monthly climatology. The authors of [17] showed that TGF-producing thunderstorms contain higher concentrations of both cloud water and ice and precipitation water and ice, which results in more vigorous lightning activity. Using the observations from the World-Wide Lightning Location Network (WWLLN) and Next Generation Weather Radars (NEXRAD), the authors of [18] found that 24 oceanic TGF-producing thunderstorms had a variety of convective strengths, ranging from relatively weak to deep convection, without distinct features. Although African thunderstorms have the highest lightning density in the world [19], they produce a relatively low TGF-to-lightning ratio in comparison with the TGF-producing thunderstorms in America and Southeast Asia, attributed by [20] to the relatively small spacing between layered charge regions in the parent thunderstorms.

Since the tropopause imposes a constraint on the vertical development of convection in thunderstorms [21], the authors of [22] found that TGFs are usually associated with a relatively high tropopause using the climatology monthly NCEP/NCAR reanalysis of tropopause altitudes for the TGF observations of the Reuven Ramaty High Energy Solar Spectroscopic Imager (RHESSI). Based on radiosonde observations, the authors of [23] recently showed that there is no considerable difference in tropopause altitudes between TGF-producing equatorial thunderstorms and those in South China (with latitude between $20°N$ and $26°N$). Considering the inter-annual variability and the significant bias in tropopause altitude from reanalysis [24,25], using tropopauses derived from high vertical resolution temperature observations associated with TGF-producing thunderstorms from Global Navigation Satellite System–Radio Occultation (GNSS–RO) would improve our understanding of the thermal structures of the environmental atmosphere associated with TGF generation. The GNSS–RO technique provides global-scale high-resolution ($\leq 0.1$ km) accurate (with error less than $1\,°C$) temperature profiles under all weather conditions [26], which is ideal for the studies of tropopause and thermal structure associated with TGF-producing thunderstorms.

To evaluate the relationship between atmospheric thermal structure and TGF genesis on a global scale, we will address the following questions in this paper: Is the tropopause associated with TGFs substantially higher than the climatology? How does the TGF-related tropopause compare over land and over ocean, and over different regions? What are the thermal structure patterns of the environmental atmosphere of TGF-producing thunderstorms? How does their intensity compare to the deep convection mentioned in previous studies [27,28] on a statistical basis?

## 2. Data and Methodology

In this work, we adopt the TGF observations from three space-borne missions: the RHESSI project from April 2006 to November 2013 [2], Fermi Gamma-ray Burst Monitor (GBM) from August 2008 to December 2019 [4,5], and Insight Hard X-ray Modulation Telescope (Insight-HXMT) from July 2017 to September 2019 [29].

We use the quality-controlled temperature profiles from the following four missions and time periods: SAC-C (Scientific Applications Satellite C, [30]) from March 2006 to August 2011, COSMIC (Formosa Satellite Mission 3/Constellation Observing System for Meteorology, Ionosphere, and Climate, [26]) from April 2006 to present, GRACE (Gravity Recovery and Climate Experiment, [31]) from March 2007 to November 2017, TerraSAR-X [32] from February 2008 to present, MetOp-A (October 2007–December 2019), and MetOp-B (February 2013–December 2019) [33–35]. All data are available from the University Corporation for Atmospheric Research (UCAR) Constellation Observing System for Meteorology, Ionosphere, and Climate (COSMIC) Data Analysis and Archive Center (CDAAC) and are provided at 0.1 km vertical resolution from the surface to about 40 km altitude. SAC-C and TerraSAR-X acquire up to 250 daily occultations (temperature profiles), MetOp-A and B provide about 630–680 daily occultations, while GRACE retrieves about 150–200 occultations and COSMIC provides up to 2500 occultations per day, respectively.

The observations from the TGF detectors and GNSS-RO are required to be within 300 km and ±3 h of each other [28], resulting in 1917 TGF-related GNSS-RO temperature profiles.

The spatial relationship between TGFs and associated lightning has been confined typically within 300 km based on the space-borne TGF detection and ground-based broadband spheric measurements [36,37]. The authors of [36] found that the majority (about 75% for TGF-associated lightning discharges located with three or more stations) of TGFs are detected within 300 km of the associated lightning, and the authors of [38] attributed half of 2188 TGFs located within 300 km range from the associated lightning discharge.

Tropopause altitudes are calculated for each GNSS-RO profile by employing the WMO temperature lapse rate tropopause definition [39], which defines the primary tropopause as the lowest altitude level at which the lapse rate is ≤2 °C km−1 and the average lapse rate between that level and any higher levels within 2 km remains less than 2 °C km−1. This definition relates directly to the static stability of the atmosphere and has been used widely in atmospheric convection studies. The WMO definition allows for additional tropopauses to be defined if, above the primary tropopause, "the average lapse rate between any level and all higher levels within 1 km exceeds 3 °C km−1, then a secondary tropopause is defined by the same criterion." The profiles that have a secondary tropopause are defined as double tropopause events [25,40]. To avoid false tropopause identification and boundary layer inversions, the algorithm is applied only to altitudes above 5 km at the pole and 10 km at the equator according to $7.5 + 2.5 \cos(2\varphi)$ km, where $\varphi$ is the latitude.

The temperature profiles are time-averaged to construct a gridded monthly climatology with a $5° \times 5°$ latitude–longitude resolution. This climatology is adopted to compare with the profiles in the vicinity of TGF-producing thunderstorms to identify anomalous temperature behavior $T_{Anomaly}$.

$$T_{Anomaly}(i) = T_{TGF}(i) - T_{Clim}(i) \tag{1}$$

where i is the altitude varying from 0 km to 39.9 km with 0.1 km vertical resolution, $T_{TGF}$ is the average temperature profile associated with TGF flashes within a $5° \times 5°$ bin, and $T_{Clim}$ is the temperature climatology for the same bin. We require these gridded temperatures, containing more than three GNSS-RO profiles.

The combined observations from the TGF-detecting satellite and GNSS-RO are illustrated in Figure 1a. An example FERMI GBM-detected TGF event (blue square) at 0916:58.3906 UTC on 7 June 2016 in the vicinity of a thundercloud region with a 230-K NCEP/CPC merged infrared (IR) brightness temperature [41] is shown in Figure 1c, with the near-simultaneous observations from GNSS-RO at 0916 UTC (red circle) and radiosonde at 1200 UTC (green diamond). The GNSS-RO temperature profile where the tropopause is identified near the altitude of 18 km is shown in Figure 1b. The coincident temperature profile and tropopause altitude from Kunming radiosonde station (66 km west of the lightning location) are superimposed, showing a tropopause at 18 km. This example shows that the tropopause remains unchanged within 300 km and 3 h.

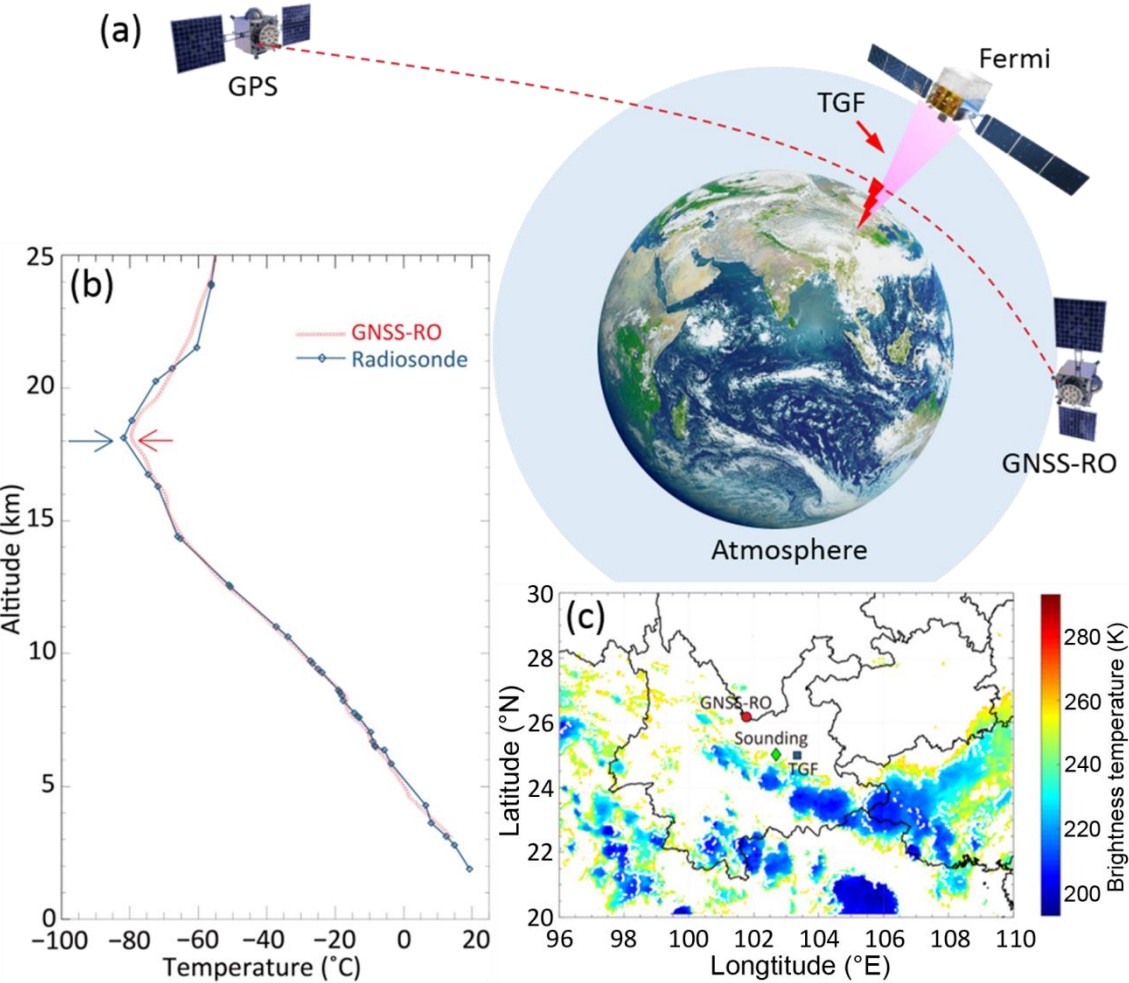

**Figure 1.** (**a**) Schematic diagram illustrating the combined observations from FERMI and GNSS-RO. (**b**) Temperature profiles from GNSS-RO at 0916 UTC (red) and radiosonde at 1200 UTC (blue) in approximation to a TGF event which occurred in Kunming, Yunnan at 0916:58.391 UTC on 7 June 2016. Colored arrows denote the locations of primary tropopause altitude calculated using each temperature profile. (**c**) NCEP/CPC merged IR brightness temperature at 0900UTC on 7 June 2016. The location of TGF event is shown by blue square, radiosonde sounding by green diamond, GNSS-RO event by red circle.

### 3. Analyses and Results

#### 3.1. TGF-Related Tropopause Altitude

The geographic distribution of the TGF-related tropopause altitude observed by GNSS-RO for four seasons from April 2006 to December 2019 is shown in Figure 2. Four seasons are labeled as DJF for December-January-February, MAM for March-April-May, JJA for June-July-August, and SON for September-October-November, respectively. In DJF and MAM, most of the TGF-related tropopauses are higher than 16.4 km. In JJA, active TGF thunderstorms concentrate in the Asian Monsoon region, Northwest Pacific, and the Caribbean Sea, with much higher tropopause altitudes (>17.6 km) in the Asian Monsoon region. This may indicate that the tropopause altitude is raised for thunderstorms. The TGF-related tropopauses show a similar geographic distribution with the climatological results [28]. In the SON season, most of the TGF-related tropopauses vary in the range from 16 to 17 km, with some lower tropopause (<15.5 km) in the Caribbean Sea and extratropics. It is noted that there are some cases of lower tropopause altitudes in Africa.

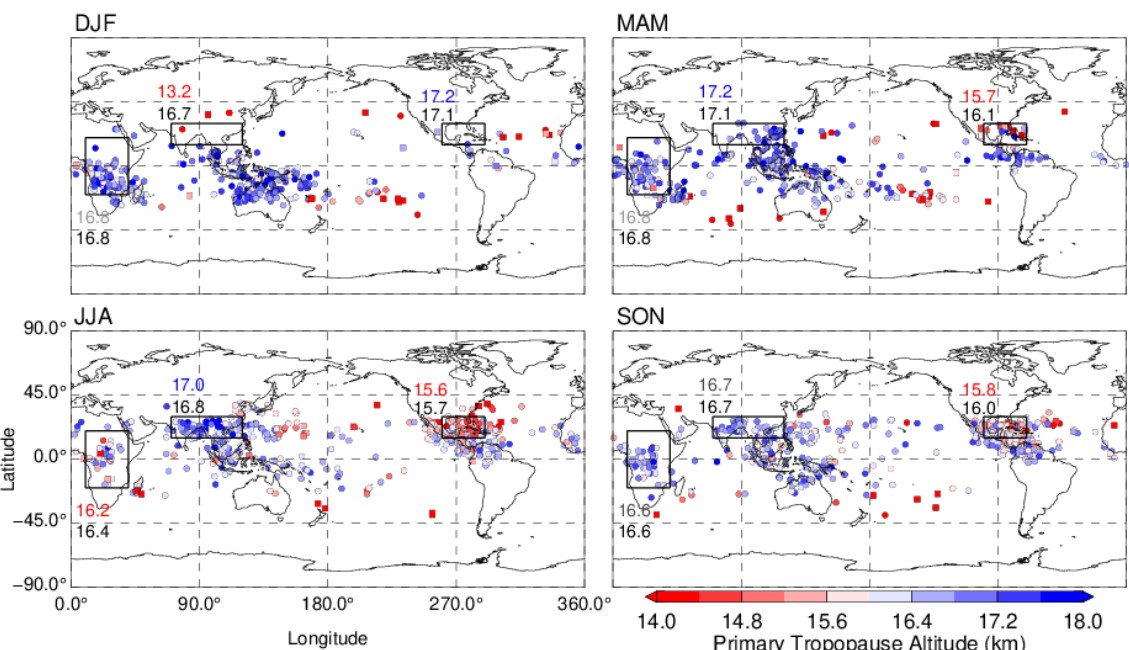

**Figure 2.** Locations of the GNSS-RO primary tropopause altitudes co-located with the TGF in a time window of 3 h and space window of 300 km for four seasons from April 2006 to December 2019. Circles denote single tropopause events and squares denote double tropopause events. The African continent, the Asian Monsoon, and the Caribbean Sea regions are depicted by rectangles. The average tropopause altitudes for the TGF events and climatology are given close to the rectangles. Red (blue) texts indicate a lower (higher) TGF tropopause than climatology (black). Gray texts indicate unexceptional values with respect to the climatology.

The regional averaged tropopause altitude for the TGF events and climatology over Africa, Asian Monsoon Anticyclone, and the Caribbean Sea is shown for each season, with blue (red) texts denoting a higher (lower) TGF-related tropopause altitude than the climatology (black). The averaged TGF-related tropopause altitude over Africa and the Caribbean Sea is 0.1–0.4 km lower than the climatology, whereas that over Asia is 0.1–0.2 km higher. The largest difference between the TGF-related and climatology tropopause occurs over Africa (−0.1 km) and Asian Monsoon Anticyclone (0.2 km) in JJA, and over the Caribbean Sea in MAM (−0.4 km).

Generally speaking, for the boreal summer season, there are two regions with active TGF productions, namely the Asian Monsoon region and the Caribbean area [22]. As we can see from JJA season in Figure 2, the average TGF-associated tropopause altitude in the Asian Monsoon region is considerably higher than that in the Caribbean (by about 1.4 km). The height of tropopause has a substantial impact on the fluences of gamma-rays detected by satellites [42,43]. If a higher tropopause is also conducive to the development of main charge layers at higher altitudes in a thunderstorm, TGFs produced by thunderstorms in the Asian Monsoon region are likely to be observed by space-born detectors.

In addition, there are some double-tropopause cases (indicated by squares in Figure 2) associated with the production of TGFs (e.g., the events in the tropical Atlantic in DJF), with incidence rates for double tropopause of 7.5%, 6.9%, and 21.4% for the NH subtropics (20° N–45° N), tropics (20° S–20° N), and the Southern Hemisphere (SH) subtropics (20° S–45° S), respectively. The incidence rate of tropical double tropopause is slightly higher than that (<5%) found by [44], while the NH subtropical example is much lower than their results (~20%).

Considering the similarity in the geographic distributions of associated TGF tropopause altitudes and climatological tropopause altitudes, we used the tropopause altitude anomaly to quantify the difference between TGF-related tropopause and the climatological examples. The tropopause altitude anomaly is defined as the monthly mean tropopause altitude associated with the TGF flashes on a 5° × 5° longitude–latitude grid minus the long-

term monthly tropopause at the same grid. We require the monthly mean TGF-related tropopause altitudes containing more than three samples. Figure 3 illustrates the seasonal cycle of the tropopause altitude anomaly for the NH and SH subtropics, as well as the tropics. Most of the median values are slightly higher than zero, which confirms the preference of TGFs for high tropopause altitudes [22]. In the tropics, some of the 5% percentiles are lower than −1 km, while most of the 95% percentiles are lower than 1 km. In the subtropics, the variability of the tropopause anomaly enlarges. In the SH subtropics, the 5% percentiles for 9 out of 12 months are lower than −2 km, whereas only 5 months have a 5%-percentile lower than −2 km in the NH. Most of the 95% percentiles for TGFs in the SH subtropics are lower than 1 km, whereas the 95% percentiles can range from 1 to 3 km in the NH subtropics. The TGF-related tropopause has a greater variability over the ocean. There is a significant seasonal cycle in the subtropics, with a smaller variability in summer and early fall and a larger one in winter and spring. In summary, most of the tropopause altitudes associated with the TGFs are slightly higher than the climatology, while some TGF-related tropopause altitudes are found to have a slightly negative bias.

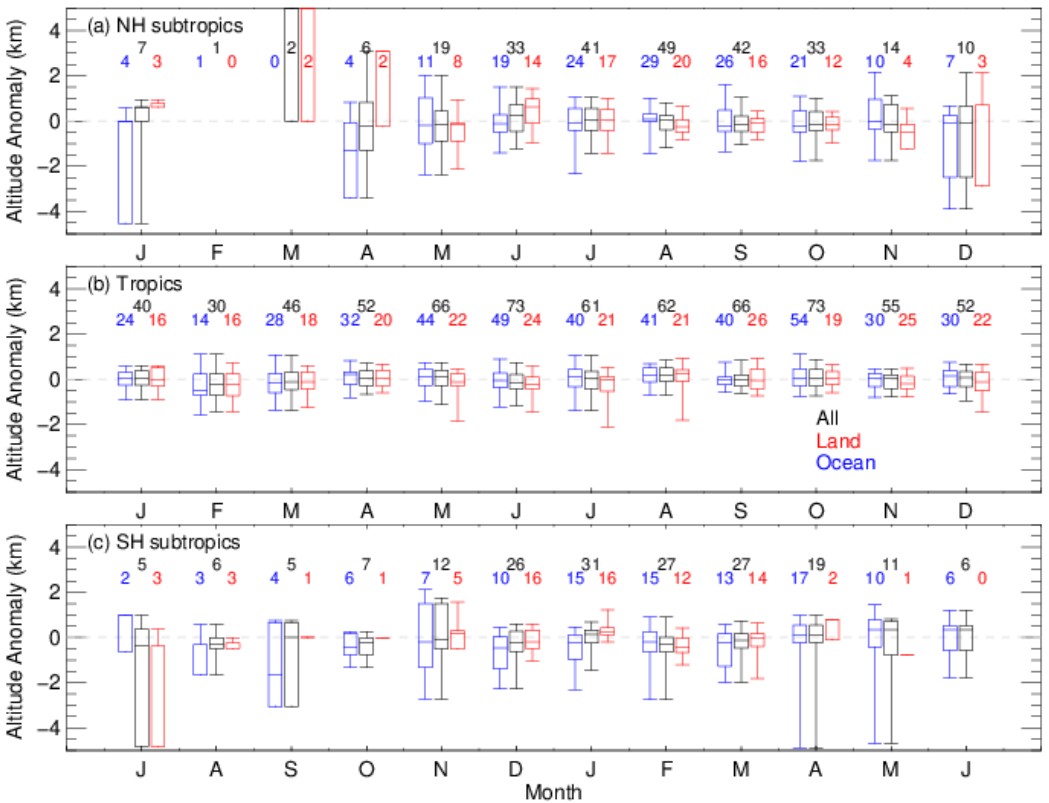

**Figure 3.** Box-and-whisker plot for monthly tropopause anomaly (km) from 2006 to 2019 for three latitude bands: (**a**) the Northern Hemisphere (NH) subtropics, (**b**) the tropics, and (**c**) the Southern Hemisphere (SH) subtropics. The vertical lines extend to the 5% and the 95% percentiles. The boxes extend from the 25% to the 75% percentiles, with a horizontal line at the median values. Blue, red and black texts indicate the grid samples of the monthly mean TGF-related tropopause altitude over the ocean, land, and globe, respectively. Note that the NH and SH annual cycles are offset by 6 months.

### 3.2. Thermal Structure of TGF-Producing Thunderstorms

To investigate the thermal structure associated with TGF production, the TGF-related tropopause structure in temperature profiles during the period of 2006–2019 for the NH extra-tropics, tropics, and SH extra-tropics over continent and ocean are shown in Figure 4. The density distribution of observed temperatures is binned in 100 m for altitude and 0.5 °C for temperature. The temperature typically decreases steadily up to about 17 km, with a sharp transition in the tropics and milder ones in the subtropics.

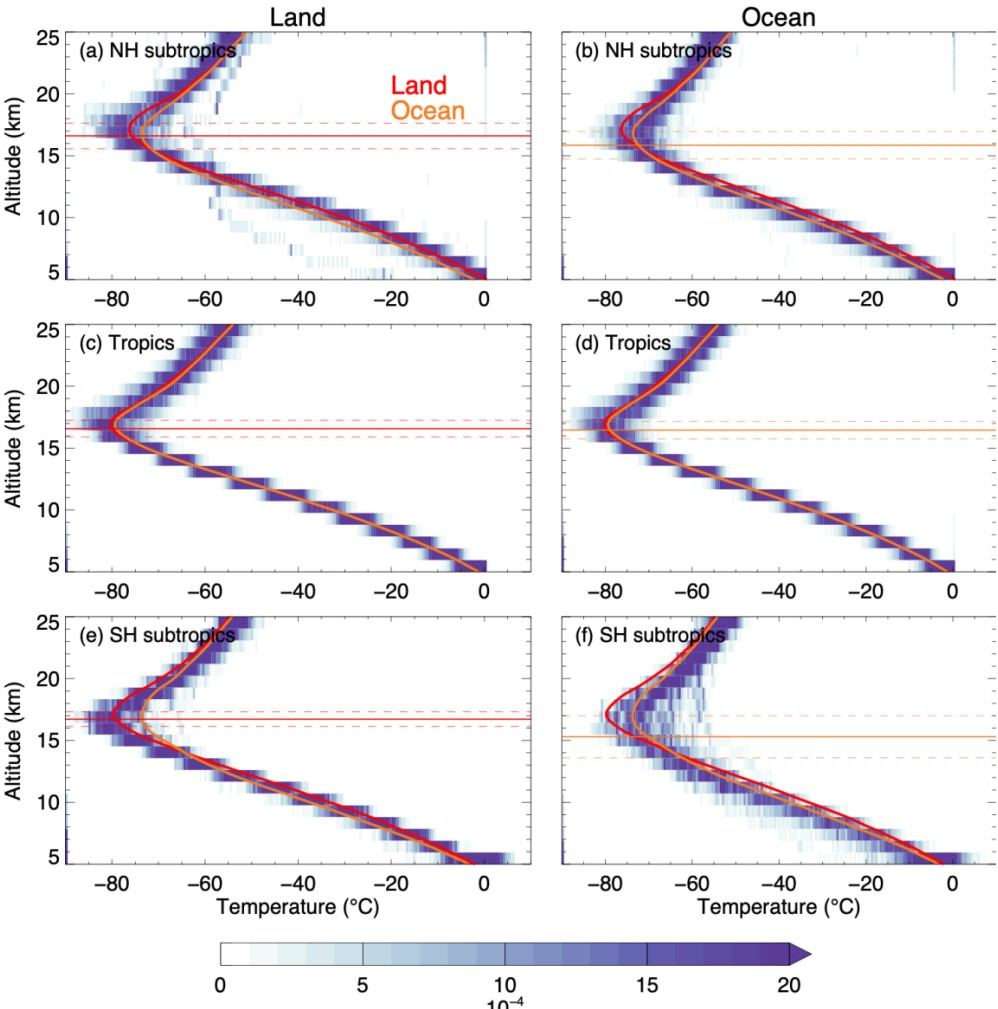

**Figure 4.** Density distribution of observed temperatures from 2006 to 2019 as a function of altitude from GNSS-RO observations co-located with TGFs (**left**: **a,c,e**) over land and (**right**: **b,d,f**) over ocean in the (top) Northern Hemisphere subtropics (20° N–45° N), (middle) tropics (20° S–20° N), and (bottom) Southern Hemisphere subtropics (20° S–45° S). In both left and right panels, the average profiles for land (red lines) and ocean (yellow lines) are superimposed. Thick horizontal lines represent the mean primary tropopause altitude in each case, with the dashed lines showing its standard deviation. Units are parts per ten thousand.

Different from previous studies which show a higher tropical tropopause altitude and a lower subtropical one [28], the tropopause altitude over continent for the tropical TGF-producing thunderstorms (16.6 km) is slightly lower than that in the subtropics (>16.6 km), while those approximating to the oceanic tropical TGFs are higher than the subtropical ones. The standard deviations of the tropopause and the corresponding temperature are about 0.7 km and 15 °C in the tropics, while those increase up to 1.1 km and 17.5 °C in the subtropics, with the highest variabilities in the SH subtropical ocean. In the tropics, there is no significant difference in the average temperature profiles between continent and ocean, but with a 1 °C colder and 0.1 km higher tropopause over land than ocean.

In the subtropics, the temperature is 2–3 °C higher in the middle and lower troposphere for the continental TGF thunderstorms, leading to a 0.5–0.6 km higher −20 °C-level and 0.8 km higher −40 °C-level than the ocean. In the tropopause region, the TGF storms have a colder and higher tropopause over land than ocean. This indicates a stronger latent heat release and a stronger cloud top outflow divergence for the continental TGF-producing storms. The largest difference in the tropopause structure between land and ocean occurs in the SH subtropics, with a 1.5 km higher and 10 °C colder tropopause.

The positive charge layer is thought to lie above the $-40\,°C$-level, while the negative charge layer generally occurs between the $-20\,°C$ and $-40\,°C$ levels [45]. The larger spacing (0.2–0.3 km) between $-20\,°C$ and $-40\,°C$ levels over land in the subtropics indicates that there might be more room for the development of negatively charged cloud region. Whether this feature will lead to a more extensive main negative cloud region in the thunderstorm merits further investigations.

The GNSS-RO temperature profiles associated with TGFs are deseasonalized by creating monthly mean profiles at each point on a $5° \times 5°$ grid and removing the long-term monthly mean profile interpolated to each profile location. With this method, we are capable of quantifying the temperature signal of TGF-producing thunderstorms shown in Figure 5, characterized by an anomalously cold middle troposphere at around 7–9 km and an anomalously cold upper troposphere at around 15 km. Some of the temperature anomaly profiles also show warm anomalies between 10–15 km, and cold anomalies between 17 and 22 km, which is consistent with the temperature signal associated with deep convection.

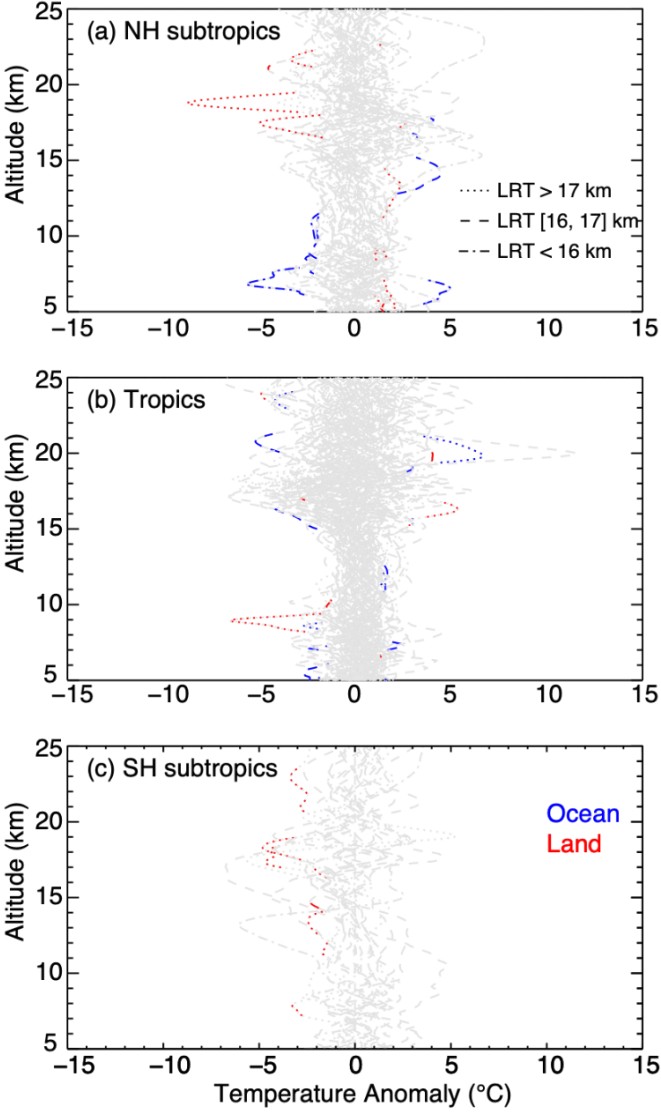

**Figure 5.** Averaged profiles of temperature anomaly within $\pm3$ h and within a radius of 300 km of a TGF event with a lapse-rate tropopause (LRT) altitude greater than 17 km (dotted lines), between 16–17 km (dashed lines), and below 16 km (dash-dotted lines) for (**a**) the Northern Hemisphere subtropics, (**b**) tropics, and (**c**) Southern Hemisphere subtropics. Red and blue curves represent continental and oceanic regions, respectively. Anomalies that exceed the 95% significance level appear in color.

Most of the TGF-related anomaly magnitudes are greater than 2 °C, with a strongest cold signal of maximum 9 °C for high tropopause in the NH extratropics (Figure 5a). The TGF-related temperature profiles tend to have larger anomalies than deep convective cloud shown in previous studies [27,28]. For example, the authors of [28] examined the temperature of tropopause-penetrating convection and found an anomalously warm upper troposphere (>2 °C) and an anomalously cold tropopause (<−3.5 °C). The colder troposphere is likely to indicate a preference for more ice in TGF-producing thunderstorms.

## 4. Discussions and Conclusions

In this study, we use the instantaneous observations from various TGF detectors (FERMI GBM, RHESSI, and *Insight*-HXMT) and GNSS-RO (SAC-C, COSMIC, GRACE, TerraSAR-X, and MetOp-A) to improve our understanding on the thermal structure associated with the production of TGFs. We used the combined observations of TGF location and temperature profile to identify the altitude of 1917 TGF-related tropopause cases. The main conclusions are summarized as follows:

1.  The TGF-associated tropopause altitudes exhibit a similar geographical distribution over the global scale and seasonal characteristics to the climatology. The regionally averaged tropopause altitude over the Caribbean Sea is 0.1–0.4 km lower than the climatology, and that over the African continent is slightly (≤0.1 km) lower than the climatology. The regional tropopause altitudes over the Asian Monsoon are 0.1–0.2 km higher than the climatology. Most of the tropopause altitudes associated with the TGFs are slightly higher than the climatology, while some of them have a slightly negative bias.
2.  There is no significant land-ocean difference in the thermal structures for the tropical TGF-producing thunderstorms. In the subtropics, the tropospheric temperature was warmed over land, with a 0.5–0.6 km higher −20 °C-level and 0.8 km higher −40 °C-level. Meanwhile, the continental TGF-producing thunderstorms have a colder and higher tropopause than the ocean.
3.  During the boreal summer season, the TGF-associated tropopause altitudes in the Asian Monsoon region are considerably higher (by about 1.4 km on average) than the Caribbean area, while whether this will lead to a higher incidence rate of TGFs relative to lightning still depends on the altitude of major charge regions of parent thunderstorms.
4.  The TGF-producing thunderstorms are characterized by a cold anomaly in the middle (around 7–9 km) and upper troposphere (around 15 km) and tend to have stronger anomalies than deep convection found in previous studies.

There is a slightly high incidence rate of double tropopause associated with the TGFs in the tropics, with some higher primary tropopause altitudes and some lower ones. The higher tropopause situations probably result from reduced stability in the lower stratosphere related to the situation of double tropopause, which may facilitate the occurrence of deep convection [46]. Meanwhile, the lower examples may result from the cooling of upper troposphere associated with the convective cloud top [28,47]. These partly indicate the relationship between deep convection and TGF occurrence, which suggests that deep convection provides favorable conditions for TGF production.

From a regional perspective, the TGF-related tropopause over the Asian Monsoon region is 0.2 km higher than the climatology for the same region. Meanwhile, the region surrounding the Tibetan Plateau has a higher altitude for the −20 °C and −40 °C levels (more than 0.3 km and ~1 km, not shown). These raised levels are associated with the Monsoon anticyclone, which is a dynamical response to the Asian monsoon convective heating from late spring to early fall [48,49]. The spacing between the −20 °C and −40 °C levels is larger over continents than ocean in the subtropics and indicates a longer distance between the layered charge regions, which accomplishes a favorable condition for TGF production [20].

In addition, the TGF-producing thunderstorms have a cooler temperature than other deep convection in the middle and upper troposphere [27,28]. The colder environment gives rise to higher concentrations of cloud ice and precipitation ice, which could be a prevalent condition conducive to TGF production [17].

The coordinated ground-based observations with respect to TGF-associated lightning are crucial for examining the physical connection between individual TGFs and their parent lightning [11,50,51]. As the continental area of South China extends to lower latitudes than the contiguous United States, the coordinated ground-based multi-instrument measurements in South China can readily obtain more desired observations to investigate the detailed connection between TGFs and parent lightning [51,52]. The Chinese Gravitational-wave high-energy Electromagnetic Counterpart All-sky Monitor (GECAM) satellite launched in December 2020 will provide more opportunities to decipher the particular physical mechanism for the production of gamma-rays [53].

**Author Contributions:** Conceptualization, T.X. and G.L.; methodology, T.X.; software, T.X. and Q.Y.; validation, T.X., J.Y. and H.Z.; formal analysis, T.X.; investigation, T.X.; resources, H.Z. and F.L.; data curation, H.Z., S.X., Q.Y. and F.L.; writing—original draft preparation, T.X.; writing—review and editing, G.L., H.Z. and J.Y.; visualization, T.X. and Y.W.; supervision, G.L.; project administration, T.X. and G.L.; funding acquisition, T.X. and G.L. All authors have read and agreed to the published version of the manuscript.

**Funding:** This work was supported by the National Key R&D Program of China (2017YFC1501501), National Natural Science Foundation of China (42075071, 41622501, and U1938115), and the Chinese Meridian Project.

**Data Availability Statement:** The TGF data are available from the online archives as follows: RHESSI (https://hesperia.gsfc.nasa.gov/hessidata/ (accessed on 10 May 2021)), FERMI (https://heasarc.gsfc.nasa.gov/FTP/fermi/data/gbm/daily/ (accessed on 10 May 2021)), and Insight-HXMT (www.hxmt.org (accessed on 10 May 2021)); the GNSS-RO data are obtained at the CDAAC (https://cdaac-www.cosmic.ucar.edu/ (accessed on 10 May 2021)), the merged IR data are from NCEP/CPC (https://doi.org/10.5067/P4HZB9N27EKU (accessed on 10 May 2021)), and the radiosonde sounding is from IGRA (https://www.ncdc.noaa.gov/data-access/weather-balloon/integrated-global-radiosonde-archive (accessed on 10 May 2021)). The *Insight*-HXMT mission is a project funded by China National Space Administration (CNSA) and Chinese Academy of Sciences (CAS).

**Conflicts of Interest:** The authors declare no conflict of interest.

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
