# Peer review of "Implications of GNSS-Inferred Tropopause Altitude Associated with Terrestrial Gamma-ray Flashes"

_remotesensing, doi:10.3390/rs13101939_

Round 1

Reviewer 1 Report

please check the pdf

Author Response

The paper under consideration is focused on the thermal structure of the environmental atmosphere associated with Terrestrial Gamma-ray Flashes (TGFs). The thermal structure is investigated with the combined observations from several detectors (FERMI, RHESSI, and InsightHXMT) and GNSS-RO (SAC-C, COSMIC, GRACE, TerraSAR-X, and MetOp-A). The geographic distributions of TGF-related tropopause altitude and climatology are similar. The regional TGFrelated tropopause altitude in Africa and the Caribbean Sea is 100-400 m lower than climatology, whereas that in Asia is 70-200 m higher. Most of the TGF-related tropopause altitudes are slightly higher than the climatology, while some of them have a slightly negative bias. The subtropical TGFproducing thunderstorms are warmer in the troposphere and have a colder and higher tropopause over land than the ocean. There is no significant land-ocean difference in the thermal structure for the tropical TGF-producing thunderstorms. The TGF-producing thunderstorms have a cold anomaly in the middle and upper troposphere and have stronger anomalies than deep convection found in previous studies.

  1. The TGF-related tropopause in Asia is typically 200 m higher than the climatology, and is considerably higher than that in Caribbean Sea.
  2. The subtropical oceanic TGF-related tropopause is lower and therefore warmer than that over continents.
  3. TGF-producing thunderstorm has (instead of have)a cooler temperature than ordinary deep convection, which likely gives rise to more cloud/precipitation ice (misprint)

I have no categorical comments on the text. All obtained results are interesting and important, and the paper may be published after minor revision (and responses to my questions). Nevertheless, it should be noted a few points that deserve the attention of the Authors:

  1. There are some misprints in the text, e.g., in the Abstract (RHESSI – misprint).

Answer:Sorry for the typo. We have corrected this in the Abstract, please see line 26 in the revised manuscript.

  1. There is no mention on the ground-based observations. Why? I mean, e.g., Armenian research group by Chilingarian et al. (2019); American research group by Rasha Abbasi et al. (2019) and some others. The results to be compared and slightly reviewed/commented in some way.

Answer:Thank you for the constructive suggestion. We have  made some revisions by adding some discussions in the Introduction part, and also including three more papers relevant.

  1. Also to be corrected: thunderstorm has (instead of have) – see above, point 3 in TGFproducing….

Answer:Sorry for the typo. We have corrected this in point 3.

  1. As to the TGFs genesis on a global scale, what is the Authors’ real contribution? According to the Authors, what questions from the list below did they manage to answer?

To evaluate the relationship between atmospheric thermal structure and TGF genesis on a global scale, we will address the following questions in this paper: Is the tropopause associated with TGFs substantially higher than climatology? How does the TGF-related tropopause compare over land and over ocean, and over different regions? What are the patterns of thermal structures of the environmental atmosphere of TGF-producing thunderstorms? How does their intensity compare to the deep convection mentioned in previous studies (e.g., Paulik and Birner, 2012; Xian and Fu, 2015) on a statistical basis?

Answer: Our study has answered the questions we raised in the Introduction and summarized them in Section 4:

Is the tropopause associated with TGFs substantially higher than climatology?
Answer: Most of the tropopause altitudes associated with the TGFs are slightly higher than the climatology, while some of them have a slightly negative bias.

How does the TGF-related tropopause compare over land and over ocean, and over different regions?
Answer: The continental TGF-producing thunderstorms have a colder and higher tropopause than the ocean. The regionally averaged TGF-associated tropopause altitudes over the Caribbean Sea and African continent are slightly lower than the climatology, which has a lower climatology tropopause altitude than the rest of the world at the same latitude. In contrast, the TGF-associated tropopause altitudes over the Asian Monsoon are slightly higher than the climatology, which has a higher climatology tropopause altitude than the other subtropical regions. 

What are the patterns of thermal structures of the environmental atmosphere of TGF-producing thunderstorms? How does their intensity compare to the deep convection mentioned in previous studies (e.g., Paulik and Birner, 2012; Xian and Fu, 2015) on a statistical basis?

Answer: The TGF-producing thunderstorms are characterized by cold anomaly in the middle (around 7-9 km) and upper troposphere (around 15 km) and tend to have stronger anomalies than deep convection found in previous studies.

Reviewer 2 Report

This is an interesting paper with a good inroduction and interesting results.

Moreover it is well written with a clear structure.

I have some editorial comments:

Page 2, line 28-33: Something or a word seems to be missing in this sentence for a complete understanding. Maybe you will like to rephrase it.

Page 3, line 18: Is „sferic“ correct?

Page 3, line 39-44:  TAnomaly, TTGF, TCLIM  are not placed in line within the text.

Figure 1: The picture could be sharper

Author Response

Page 2, line 28-33: Something or a word seems to be missing in this sentence for a complete understanding. Maybe you will like to rephrase it.

Answer: This sentence has been modified as “Considering the inter-annual variability and the significant bias in tropopause altitude from reanalysis (Son et al., 2011; Xian and Homeyer, 2019), using tropopauses derived from high vertical resolution temperature observations associated with TGF-producing thunderstorms from Global Navigation Satellite System-Radio Occultation (GNSS-RO) would improve our understanding on the thermal structures of the environmental atmosphere associated with TGF generation.” Please see Lines 76-82 in the revision.

Page 3, line 18: Is „sferic“ correct?

Answer: Yes, sferic measurements represent the measurements of radio frequency lightning emissions.

Page 3, line 39-44:  TAnomaly, TTGF, TCLIM  are not placed in line within the text.

Answer: Necessary changes have been made to Lines 139-145 in the revision.

Figure 1: The picture could be sharper

Answer: Thank you for the suggestion. We have reedited this figure.

Reviewer 3 Report

The paper "Implications from GNSS-inferred tropopause altitude associated with Terrestrial Gamma-Ray Flashes", by Xian et al., investigates the environmental conditions favourable to TGF occurrence. In particular, the thermal structure of the troposphere and the height of the tropopause, as detected by GNSS-RO data, is studied in comparison with TGF catalogues from orbiting detectors (RHESSI, FERMI, Insight-XMT).

The topic is rather novel and the approach is original and interesting, the paper is well written and in my opinion it deserves publication on Remote Sensing. However, I think than some deeper discussion is needed before publication, possibly following my suggestions below. 

First: please add linenumbers. It is very difficult to provide a detailed review without the possibility to refer to the text.

3rd highlights: what does exactly mean "...thunderstorm have a cooler temperature than ordinary deep..."? what is the temperature of a thunderstorm? is the thunderstorm isothermal? By the way: temperature is lower not cooler.

line 3. Write RHESSI instead of RHESS

Introduction. The CAPE issue was recently considered with similar approach also in Tiberia et al (2021) (doi.org/10.3390/rs13040784).

beginning of page 4: "This example confirms that the tropopause remains unchanged within 300 km and 3 hours." This sentence is rather superficial: one example is reported and no guess on what happens in general is possible. This is not a validation (see caption of figure 1). 

last line of the caption of figure 1: the colors of the symbols do not match the description.

Section 3.1. Here in the text the estimated height of the troposphere is implicitly given with the precision of 100 meter (e.g. 17.6 km) as it is also reported in section 2, while in figure 2 the tropopause height has two decimal digits (i.e, 10m precision). Moreover, the differences in height are given in meters on page 5. Please, decide the precision of your estimate and stick to this throughout the paper.

How, and why, are the three areas (Africa, Caribbean and Monsoon) selected? They have different extension, aspect ratio and background. This choice should be justified.

the sentence "This may indicate that the tropopause altitude is raised for thunderstorms" is not clear. Are TGF-thunderstorms or all thunderstorms? Are thunderstorms modifying the tropopause, or vice versa?

What is the meaning of the height intervals 100-400 m and 70-200? Are they min-max or what? how are distributed the height differences within the intervals?

Does this section reply to the question posed at the beginning "Is the tropopause associated with TGFs substantially higher than climatology?"? It seems the answer is no.

Section 3.2. in the sentence "The variabilities of the tropopause and temperature profiles are about 0.7 km and 15°C in the tropics, while those increase up to 1.1 km and 17.5°C...", what are the "variabilities"? standard deviation? Moreover: what is the "variability of temperature profiles" given as a single value?

the sentence "The larger spacing between -20°C and -40°C levels over land in the subtropics indicates a longer distance between the charge centers in the continental storms." is highly questionable. The fact that -20°C and -40°C isotherms are more distant (how much? 200 to 400 meters?) does not imply more distance between positive and negative centers: it only tells that the position of negative center is more undefined, or it spans over wider depth.

end of pag. 9. The relationship between "colder troposphere" (at some levels) and the amount of ice in the cloud is not so obvious as the Authors suppose: many complex mechanisms are responsible for ice formation in a thundercloud, and are not considered here.  

Conclusion last sentence: "... satellite to be launched at the end of 2020 ..." Is now GECAM flying? Please, update.

Author Response

First: please add linenumbers. It is very difficult to provide a detailed review without the possibility to refer to the text.

Answer: Linenumbers have been added to the revision.

3rd highlights: what does exactly mean "...thunderstorm have a cooler temperature than ordinary deep..."? what is the temperature of a thunderstorm? is the thunderstorm isothermal? By the way: temperature is lower not cooler.

Answer: This highlight have been modified into “TGF-producing thunderstorms have stronger cold anomalies than ordinary deep convection, which likely gives rise to more cloud/precipitation ice.” Please see Lines 44-45 in the revision.

line 3. Write RHESSI instead of RHESS

Answer: Sorry for the typo. Corrected.

Introduction. The CAPE issue was recently considered with similar approach also in Tiberia et al (2021) (doi.org/10.3390/rs13040784).

Answer: Necessary changes have been made in Lines 56-57, and a reference has been added to the revision. Thank you for the information.

beginning of page 4: "This example confirms that the tropopause remains unchanged within 300 km and 3 hours." This sentence is rather superficial: one example is reported and no guess on what happens in general is possible. This is not a validation (see caption of figure 1). last line of the caption of figure 1: the colors of the symbols do not match the description.

Answer: This sentence has been modified into “This example shows that the tropopause remains unchanged within 300 km and 3 hours.” Please see Lines 156-157 in the revision. Necessary changes have been made in the caption of figure 1.

Section 3.1. Here in the text the estimated height of the troposphere is implicitly given with the precision of 100 meter (e.g. 17.6 km) as it is also reported in section 2, while in figure 2 the tropopause height has two decimal digits (i.e, 10m precision). Moreover, the differences in height are given in meters on page 5. Please, decide the precision of your estimate and stick to this throughout the paper.

Answer: Thank you for the constructive suggestion. We use 0.1 km precision for our estimated tropopause throughout the revision.

How, and why, are the three areas (Africa, Caribbean and Monsoon) selected? They have different extension, aspect ratio and background. This choice should be justified.

Answer: The observed TGFs are sparsely clustered in these three regimes, which are not selected for quantitatively inter-comparison among regimes. Instead, we examine the tropopause altitude for each regime to understand the difference between climatology and TGF-producing environment.

the sentence "This may indicate that the tropopause altitude is raised for thunderstorms" is not clear. Are TGF-thunderstorms or all thunderstorms? Are thunderstorms modifying the tropopause, or vice versa?

Answer: The tropopause altitude is raised for TGF-producing thunderstorms. Necessary changes have been made to this sentence. Please see Line 169 in the revision. According to previous studies (e.g., Xian and Fu, 2015), only the tropopause-penetrating convection can lead to a rapid (within 20 min) lift of the tropopause by the adiabatic lofting within the convection (within a 10 km radius).  However, most of our samples lie out of this small spatial-temporal scale. The higher tropopause here provides a deeper extent for TGF-producing thunderstorm development.

What is the meaning of the height intervals 100-400 m and 70-200? Are they min-max or what? how are distributed the height differences within the intervals?

Answer: They are minimum and maximum, indicating the range of the differences for a region. We explained the difference distribution in the following sentence. Please see Lines 179-181 in the revision.

Does this section reply to the question posed at the beginning "Is the tropopause associated with TGFs substantially higher than climatology?"? It seems the answer is no.

Answer: We posed a question in the Introduction and found that most of the TGF-related tropopause altitudes are slightly higher than the climatology.

Section 3.2. in the sentence "The variabilities of the tropopause and temperature profiles are about 0.7 km and 15°C in the tropics, while those increase up to 1.1 km and 17.5°C...", what are the "variabilities"? standard deviation? Moreover: what is the "variability of temperature profiles" given as a single value?

Answer: Necessary changes have been made to this sentence. Please see Lines 230-233 in the revision.

the sentence "The larger spacing between -20°C and -40°C levels over land in the subtropics indicates a longer distance between the charge centers in the continental storms." is highly questionable. The fact that -20°C and -40°C isotherms are more distant (how much? 200 to 400 meters?) does not imply more distance between positive and negative centers: it only tells that the position of negative center is more undefined, or it spans over wider depth.

Answer: Thank you for the constructive comments. The larger spacing (0.2-0.3 km) between -20°C and -40°C levels over land in the subtropics indicates that there might be more room for the development of negatively charged cloud region. Whether this will lead to a more extensive main negative cloud region in the thunderstorm merits further investigations. Necessary changes have been made in Lines 245-249 in the revision.

end of pag. 9. The relationship between "colder troposphere" (at some levels) and the amount of ice in the cloud is not so obvious as the Authors suppose: many complex mechanisms are responsible for ice formation in a thundercloud, and are not considered here.  

Answer: Indeed, both the upward motion (mainly depends on large-scale circulation) and radiative cooling (depends on the concentrations of water vapor and aerosol) are responsible for ice formation in a thunderstorm. Here we only focus on the thermal structure of TGF-producing thunderstorms and find a strong cold anomaly, which implies a preference for ice formation. Necessary changes have been made in Lines 265-266 in the revision.

Conclusion last sentence: "... satellite to be launched at the end of 2020 ..." Is now GECAM flying? Please, update.

Answer: Necessary changes have been made to this sentence. Please see Lines 322-325 in the revision.

Round 2

Reviewer 3 Report

The authors have satisfactorily responded to all my comments and made
the necessary changes to the manuscript, that now can be published on
Remote Sensing.
I only suggest few more minor changes:
Caption of Figure 1: please, remove the first sentence: this is not a validation.
Figure 5: please define in the text what LRT means.

Author Response

The authors have satisfactorily responded to all my comments and made the necessary changes to the manuscript, that now can be published on Remote Sensing.
I only suggest few more minor changes:
Caption of Figure 1: please, remove the first sentence: this is not a validation.

Answer: Thanks. We have removed the first sentence in Caption of Figure 1.

Figure 5: please define in the text what LRT means.

Answer: We have added the description of LRT to the Caption.